# Characterisation of biomarkers of intestinal barrier function in response to a high fat/high carbohydrate meal and corticotropin releasing hormone

**Tamara Mogilevski** [ID] *, **Sam Rosella, Anke Nguyen, Jessica Fitzpatrick, Francis Parker, Emma P. Halmos** [ID]**, Peter R. Gibson**

Department of Gastroenterology, Central Clinical School, Monash University and Alfred Health, Melbourne, Australia

* tamara.mogilevski@monash.edu

## Abstract

### Background

Variation of circulating concentrations of putative biomarkers of intestinal barrier function over the day and after acute physiological interventions are poorly documented on humans. This study aimed to examine the stability and pharmacokinetics of changes in plasma concentrations of intestinal Fatty-acid -binding -protein (IFABP), Lipopolysaccharide-binging–protein (LBP), soluble CD14, and Syndecan-1 after acute stress and high fat-high-carbohydrate meal.

### Methods

In a single-blinded, cross-over, randomised study, healthy volunteers received on separate days corticotropin-releasing hormone (CRH, 100 µg) or normal saline (as placebo) intravenously in random order, then a HFHC meal. Participants were allowed low caloric food. Markers of intestinal barrier function were measured at set timed intervals from 30 minutes before to 24 hours after interventions.

### Results

10 participants (50% female) completed all three arms of the study. IFABP decreased by median 3.6 (IQR 1.4–10)% from -30 minutes to zero time (p = 0.001) and further reduced by 25 (20–52)% at 24 hours (p = 0.01) on the low caloric diet, but did not change in response to the meal. Syndecan-1, LBP and sCD14 were stable over a 24-hour period and not affected acutely by food intake. LBP levels 2 hours after CRH reduced by 0.61 (-0.95 to 0.05) µg/ml compared with 0.16 (-0.3 to 0.5) µg/ml post placebo injection (p = 0.05), but other markers did not change.

### Conclusion

Concentrations of IFABP, but not other markers, are unstable over 24 hours and should be measured fasting. A HFHC meal does not change intestinal permeability. Transient reduction of LPB after CRH confirms acute barrier dysfunction during stress.

**Data Availability Statement:** All relevant data are within the paper and its Supporting Information files.

**Funding:** The authors received no specific funding for this work.

**Competing interests:** The authors have declared that no competing interests exist.

## Introduction

The last decade has seen a number of studies supporting the theory of chronic intestinal barrier perturbation in the pathogenesis of many human diseases, with inflammatory bowel disease (IBD) having the greatest degree of evidence for this association [1–4]. Acute and chronic stress has been implicated in the propagation of inflammation in patients with IBD [5–7]. The mechanism proposed for this phenomenon includes the ability of stress to induce a breakdown in intestinal barrier function via mast cell- and corticotropin-releasing hormone (CRH)-dependent pathways [8]. In addition, diets rich in ultra-processed and fried foods, are associated with the development of Crohn's disease [9–11]. Intestinal barrier perturbation, in the setting of stress via psychological and dietary challenges, have been demonstrated experimentally in humans using stress paradigms and dietary manipulation with variable success [12–16]. Replicating stress paradigms brings practical difficulties in a research setting, whilst dietary manipulation studies have produced mixed outcomes with their effect on barrier function. (11–14) Intestinal permeability changes similar to those observed with stress paradigms can be mimicked by the peripheral administration of CRH [17, 18].

Several non-invasive markers have been proposed as useful tools in measuring intestinal permeability changes in vivo. Those readily measurable in blood samples include lipopolysaccharide-binding protein (LBP), soluble CD-14 (sCD14), syndecan-1 (CD138) and intestinal-type fatty acid-binding protein (IFABP). sCD14 and LBP are part of the lipopolysaccharide (LPS) inflammatory signalling pathway, their circulating levels and ratio (LBP:sCD14) are, therefore, considered to be objective markers of LPS translocation [15, 19]. IFABP is an intracellular protein expressed in the epithelial cells of predominantly the small intestine. It is involved in intracellular nutrient processing and is prone to leakage into the bloodstream from enterocytes in the setting of small intestinal damage [20, 21]. Syndecan-1 (CD138) is a ubiquitous cell-surface protein which forms a key component of the glycocalyx layer present on all adherent and many non-adherent cells [20]. It is crucial to the maintenance of epithelial—including intestinal epithelial—barrier function [22]. Its proteolytic cleavage into the circulation has been associated with the perturbation of the intestinal barrier, but also with generalised inflammation [23–25].

Potentially confounding physiological factors that might influence the concentrations of these molecules have received only limited attention. Issues vary from the simple–such as diurnal variation, stability over time and the effect of food ingestion–to the more complex—IFABP and syndecan-1 are preformed and, therefore, can be rapidly released on intestinal injury, whereas the timeline of changes in circulating markers, LPB and sCD14—which can be mopped up and then synthesis induced on exposure to LPS—require critical examination.

In this single-blinded, placebo-controlled crossover study, we aimed first, to establish the diurnal variation of the aforementioned plasma biomarkers of intestinal permeability and second, to evaluate the performance of these biomarkers and the optimal timing of blood sampling in two models previously claimed to induce acute perturbation of intestinal permeability perturbation in healthy human volunteers; acute stress associated with intravenous corticotropin-releasing hormone (CRH) [17, 18] and the consumption of a high-fat high-carbohydrate (HFHC) meal (11–14). A cross-over design was used with each putative inducer of change in intestinal permeability with a common placebo arm in order to minimise inter-individual variability that may be considerable in such measures.

## Methods and materials

### Participants

Healthy subjects, 5 male and 5 female aged between 18 and 65 years, were recruited from affiliated members of Monash University and the Alfred Hospital between December 2020 and

October 2021. Exclusion criteria included the presence of IBD, coeliac disease, a diagnosis of irritable bowel syndrome or significant gastrointestinal symptoms not otherwise medically diagnosed, type 1 or type 2 diabetes, use of non-steroidal anti-inflammatory medications, probiotics or antibiotics 2 months prior to study entry, pregnancy, breastfeeding, and allergies or intolerances to any of the proposed meals. 10 healthy participants were recruited for this study–a number similar to previously published data in this field. (13–19)

## Study protocol

This was a randomized, placebo controlled, cross-over trial in healthy humans. A flowchart of the study procedures can be found in (Fig 1). After assessment by the study doctor to ensure eligibility and seek written, informed consent, the participants were given a list of foods (a snack and dinner menu) low in fat and carbohydrates (LFLC). These were the only foods that they were able to consume during the study. Participants were also asked to abstain from alcohol and intense exercise for 3 days prior to each of the study visits.

Participants attended their first visit in the morning after an optional light breakfast (consisting of LFLC food options) and were subsequently randomised to receive either a CRH or placebo injection first. This was calculated by using randomly computed blocks with random block sizes generated by the website www.randomisation.com. They had a cannula inserted and blood taken. This blood draw was designated as t = -30 min, Participants had a further blood test at the t = 0 mark and then had either 100 µg dose of CRH (Ferring, Kiel, Germany) or placebo (10ml of normal saline), both given as a bolus slowly via the cannula over 30 seconds. They then had further blood tests after 2 and 6 hours as they remained near the sampling site. After t = 6 hours, the cannula was removed. A LFLC late lunch meal was provided to the

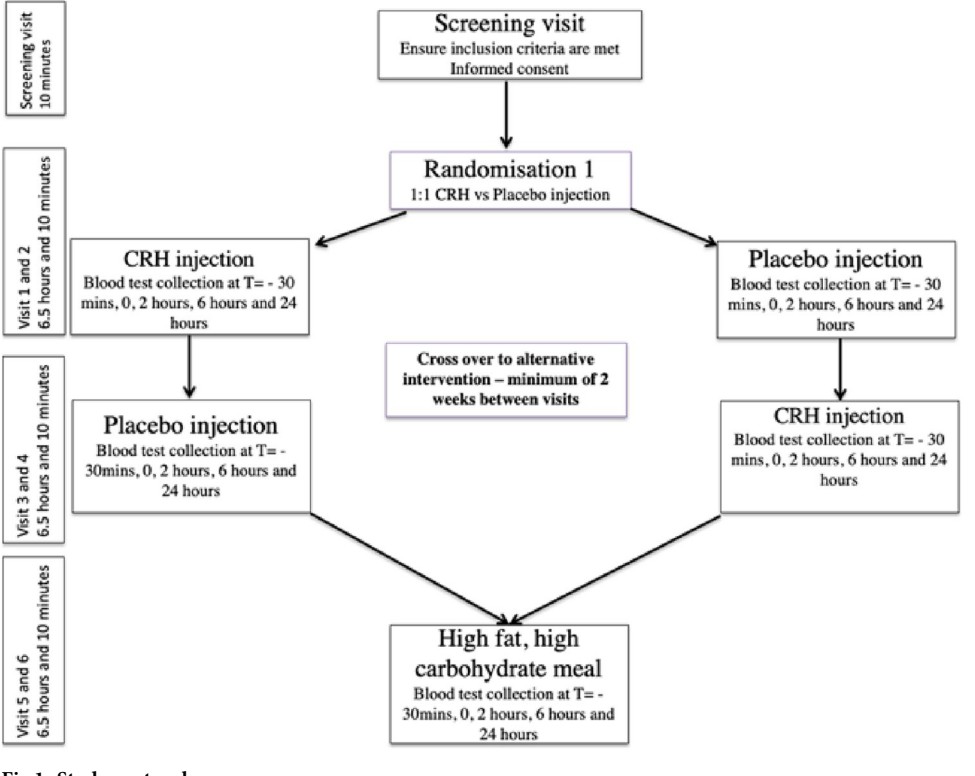

**Fig 1. Study protocol.**

participants on completion of this visit and instructions preparing a LFLC dinner for that evening were reinforced. The participant returned 24 hours after the initial CRH or placebo injection in order to have their final blood test. These time points were chosen based on previous studies [17, 26] and were thought to be most representative of the changes seen in intestinal permeability biomarkers over a 24-hour period. Again, they were permitted a light breakfast consisting of LFLC foods prior to the final blood test. After at least two weeks' washout, participants returned and followed the same protocol except that the alternate injection was given. Compliance with the LFLC diet was evaluated by specific questioning on the morning of visit 1.

After another minimum 2-week washout period, participants returned and the same protocol was followed except that the intervention was a standardised HFHC meal (in place of the CRH or placebo injection), as had been applied in other studies [12, 13, 26]. This comprised two sausage and egg muffins and two hash browns (McDonalds, South Yarra, Victoria, Australia). The macronutrient content was 4314kJ energy, 45.8 g protein, 56 g total fat, 82.2 g carbohydrate (McDonalds website: https://mcdonalds.com.au/maccas-food/nutrition, as at December 2019). All participants were given the HFHC meal as the last intervention—this part of the protocol was not, and unable to be, blinded. Compliance with meal consumption was checked by the study doctor conducting the experiment.

The protocol was approved by the Monash University Research Ethics Committee (approval number 226/37) and registered as a randomised controlled trial at the Australian New Zealand Clinical Trials Registry (ACTRN12622000730707). Registration was inadvertently overlooked until after the study had commenced, but no changes to the study protocol were made between ethics approval and trial registration. The authors confirm that all ongoing and related trials for this drug/intervention are registered.

## Biomarker analyses

Peripheral venous blood samples were collected into EDTA tubes [15, 18]. Plasma was extracted after centrifugation and stored in aliquots at -80˚C until assayed. Concentrations of the biomarkers were measured by commercially available enzyme-linked immunosorbent assays (ELISA) on freshly thawed plasma. All samples from an individual were assayed on the same ELISA plate. All were performed according to manufacturers' protocols. The assay kits were for human IFABP (R&D Systems, USA), LBP (Hycult Biotech, The Netherlands), human sCD14 (R&D Systems, USA), and human syndecan-1 (CD138) (Diaclone, France). The average coefficient of variation between duplicates was below 10%. Averages of duplicates were determined and absolute values are expressed in the following units: pg/mL for IFABP, µg/mL for LBP and ng/ml for sCD14 and syndecan-1. The LBP:sCD14 ratio was calculated by converting sCD14 from ng/ml to µg/mL and dividing the LBP concentration by the sCD14 concentration.

## Statistical analysis

As this was an exploratory study without previous data to identify predicted variance, power calculations were not performed. A pragmatic number of subjects was determined together with a cross-over design to minimise confounding. As this study was performed during the COVID-19 pandemic, recruitment was limited to subjects working within the hospital precinct. Hence, the aim was to analyse 10 subjects with complete data, where the subjects acted as their own control.

Data are presented as median (IQR) or mean (SEM) depending on normality of data distribution. Differences across different time points were evaluated by repeated-measures ANOVA

or Friedman's test and between-time points using the Wilcoxon signed rank test or paired t-test where appropriate. Bonferroni correction was applied when multiple points were measured. Hence, for analysis of the two acute time points from baseline to 6 hours, a p-value $\leq 0.025$ was considered statistically significant. The same placebo data (associated with normal saline injection) were used for each intervention, which were compared with the intervention results separately. The baseline concentration of the biomarkers was considered to be the level taken just prior to intervention (i.e., at t = 0). Since there was some variance in t = 0 values across the interventions and the aim of the study was to determine temporal differences, all values for the active arms were corrected by the difference between that arm and the placebo at t = 0. The level of statistical significance was set at $p \leq 0.05$ except where correction was made for multiple comparisons. All statistical analyses were performed using Prism V8.30 (GraphPad Software LLC).

## Results

### Participants

Ten healthy volunteers with a median age of 36 (range 26–63) years were recruited. One half were female and body mass index was 22.6 (21.5–25.0) kg/m$^2$. 7/10 participants had a LFLC breakfast 30 to 60 minutes prior to their initial blood test for all studies. One fasted for all and two fasted prior to the CRH, but not placebo or food intervention study visits. All participants adhered to the LFLC diet during the relevant time periods. The CRH injection and HFHC meal were tolerated well with no significant adverse effects reported.

### Diurnal variations in biomarker concentrations

Variations of the concentrations of biomarkers over a 24-hour period were examined where saline injection was the only intervention (Fig 2). Plasma IFABP levels differed across the time points (p = 0.001; Friedman's test). Between-time point analysis showed that concentrations reduced by 25 (IQR 20–52)% 24 hours after placebo compared with baseline (p = 0.01). Differences between the other time points were not statistically significant. In contrast, other biomarkers were stable (no differences across or between time points) over the 24 hours.

Before all interventions, 2 blood samples were taken to determine the stability of the biomarker during the immediate baseline period. These analyses, were performed using a combination of all t = -30 and t = 0 samples for each biomarker (n = 30) (Fig 3). Again, IFABP levels were not stable, with a median 3.6 (1.4–10)% reduction of concentrations over the 30 minutes (p = 0.001; Wilcoxon ranked sign test). No differences were observed for the other biomarkers.

### The acute effect of CRH on biomarker concentrations

In order to establish the influence of CRH on the biomarkers acutely, the pattern of results over 6 hours were analysed across the 2- and 6-hour time points. Differences between the individual time points were examined, for which P<0.025 was considered statistically significant after Bonferroni's correction.

Concentrations of IFABP (p = 0.2 ANOVA), syndecan-1 (p = 0.8 ANOVA) LBP:sCD14 (p = 0.4, Friedman) and sCD14 (p = 0.4) and did not differ from those following saline injection over 6 hours, but LBP did (p = 0.03) (Fig 4). Differences between individual time points were not statistically significant for any of the time points or biomarkers measured. Plasma LBP reduced at 2 hours after CRH injection compared with that after the saline injection; (p = 0.05, Wilcoxon), but this was not statistically significant after correction for multiple comparisons.

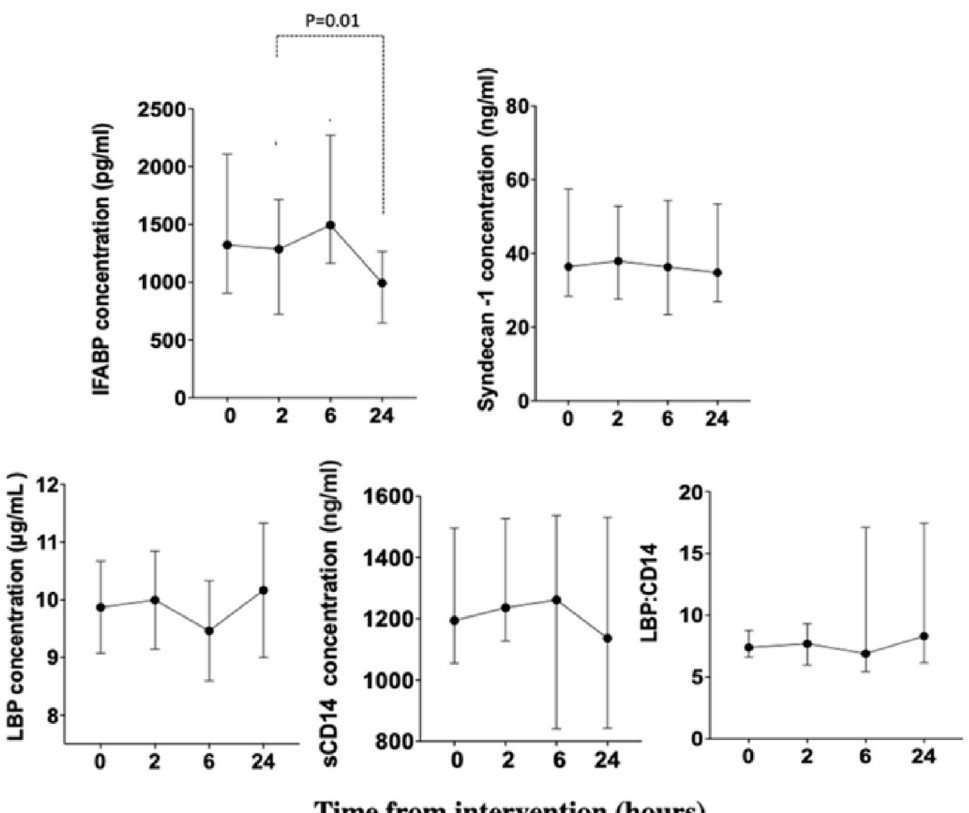

**Fig 2. Concentrations of biomarkers—intestinal fatty acid-binding protein (IFABP), syndecan-1, lipopolysaccharide-binding protein (LBP), soluble CD14 (sCD14) and LBP:sCD14 ratio—over 24 hours with a low fat, low carbohydrate meal plan after the placebo (saline) injection.** No differences across the time points were observed for syndecan-1, LBP, sCD14 or LBP:sCD14 ratio (ANOVA or Friedman's test). On pair-wise comparisons, levels of IFABP showed statistically significant differences as indicated (Wilcoxon). Data are shown as mean with standard error of mean for LBP and median with inter-quartile ratios for all other biomarkers.

## The acute effect of the HFHC meal on biomarker concentrations

As with CRH, the pattern of results over the 6 hours were then analysed across the 2- and 6-hour time points in order to establish the influence of the HFHC meal on the biomarkers acutely. Differences between the individual time points were examined, for which P<0.025 was considered statistically significant after Bonferroni's correction.

Concentrations of IFABP differed from those following saline injection over 6 hours (p = 0.02, ANOVA) whereas LBP (p = 0.4, Friedman), sCD14 (p = 0.4), syndecan-1 (p = 0.8) and sCD14/LBP (p = 0.7) did not (Fig 5) Differences between individual time points were not significant for any of the time points or biomarkers measured. However, IFABP concentrations reduced after the HFHC meal when compared to the saline injection at 6 hours (p = 0.04, Wilcoxon), but this was not statistically significant after correction for multiple comparisons.

## The effect of CRH and the HFHC meal on biomarkers at 24 hours

After correction for baseline values, concentrations of biomarkers were assessed 24 hours after placebo injection and compared to post CRH and the HFHC meal. No differences were detected between placebo and post intervention plasma biomarkers levels after either intervention (Fig 4).

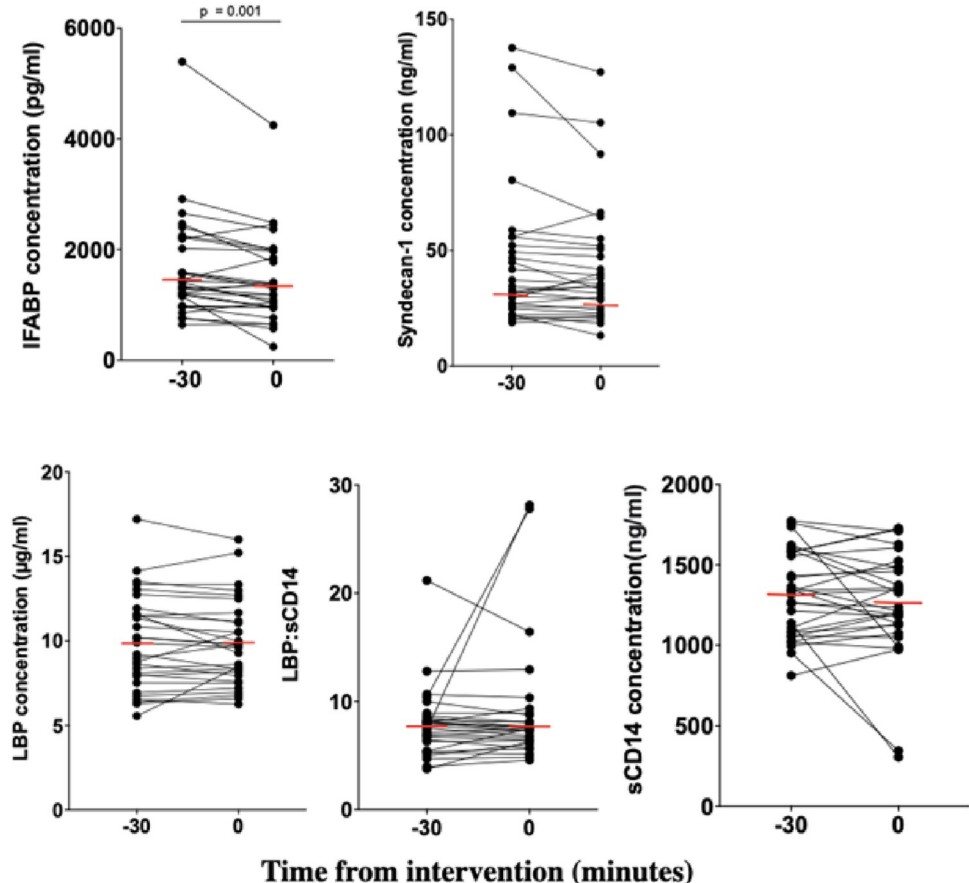

**Fig 3. Concentrations of biomarkers—intestinal fatty acid-binding protein (IFABP), syndecan-1, lipopolysaccharide-binding protein (LBP), soluble CD14 (sCD14) and LBP:sCD14 ratio—during the immediate baseline period.** All data from plasma taken 30 minutes before and at the intervention were pooled. Plasma intestinal fatty acid-binding protein (IFABP) reduced from -30 minutes to zero-time (p = 0.001; Wilcoxon). No differences were observed for syndecan-1, lipopolysaccharide-binding protein (LBP), soluble CD14 (sCD14) or LBP:sCD14 ratio. Horizontal lines (in red) represent median concentrations of each biomarker.

## Discussion

In this single-blinded, randomised, cross-over, placebo-controlled study, the variations of five proposed markers of intestinal permeability over 24 hours were documented to determine the diurnal patterns and the effect after two interventions—CRH exposure and a HFHC meal administration—that have previously been reported to cause acute changes in intestinal permeability function [12–18, 26, 27]. The results from such studies have been extrapolated to clinical implications, particularly in patients with IBD [5–7, 10, 28], without clarification in controlled environments. The key findings were first, that plasma IFABP levels are not stable over a 24-hour period, nor at the two baseline readings, taken 30 minutes apart. The highly variable levels make their interpretation in relation to effects from CRH and HFHC difficult, but the reduction after oral intake occurred to a greater degree after a HFHC meal than whilst on a low-calorie diet. Second, syndecan-1, LBP, sCD14 and the LBP:sCD14 ratio are stable over a 24-hour period and are not affected acutely by food intake. Third, the previous observation of a significant CRH-induced plasma IFABP increase [18] could not be clearly replicated in this study. Last, that LBP levels tended to be lower 2 hours after CRH injection when compared with placebo whilst there were no acute changes to the markers of LPS translocation or syndecan-1 after the HFHC meal.

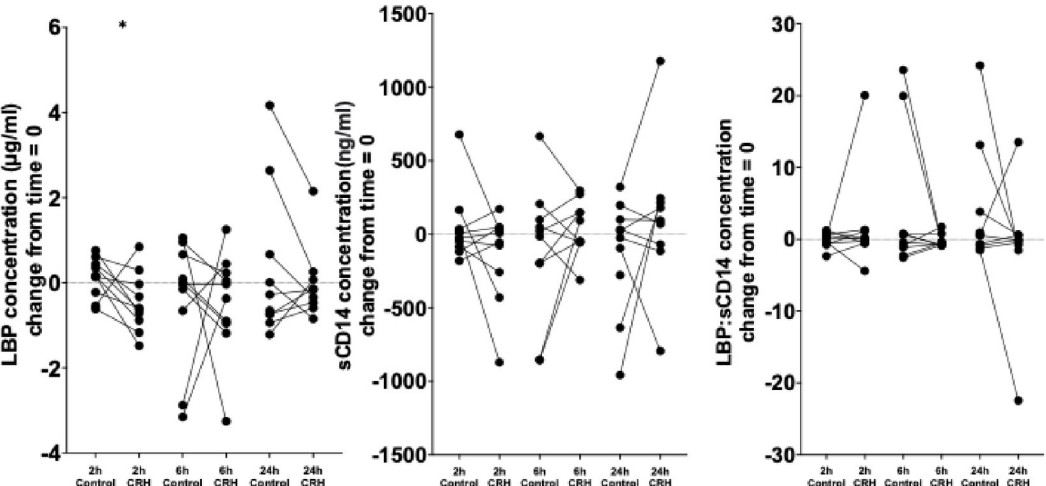

**Fig 4. Change from baseline (t = 0) in the concentrations of biomarkers—intestinal fatty acid-binding protein (IFABP), syndecan-1, lipopolysaccharide-binding protein (LBP), soluble CD14 (sCD14) and LBP:sCD14 ratio—after 2, 6 and 24 hours of 100 μg corticotropin-releasing hormone (CRH) or normal saline (Control) in a similar volume intravenously.** At the 2-h time point, LBP concentration was lower after CRH injection compared with the control (shown as an asterisk) with a post CRH LBP median level of -0.61 (IQR -0.95 to 0.05) μg/ml) vs 0.16 (-0.3 to 0.5 ug/ml) post saline injection (p = 0.05; Wilcoxon), although statistical significance was lost if the data were corrected for multiple comparisons. No other differences for biomarkers at any of the other time points were identified.

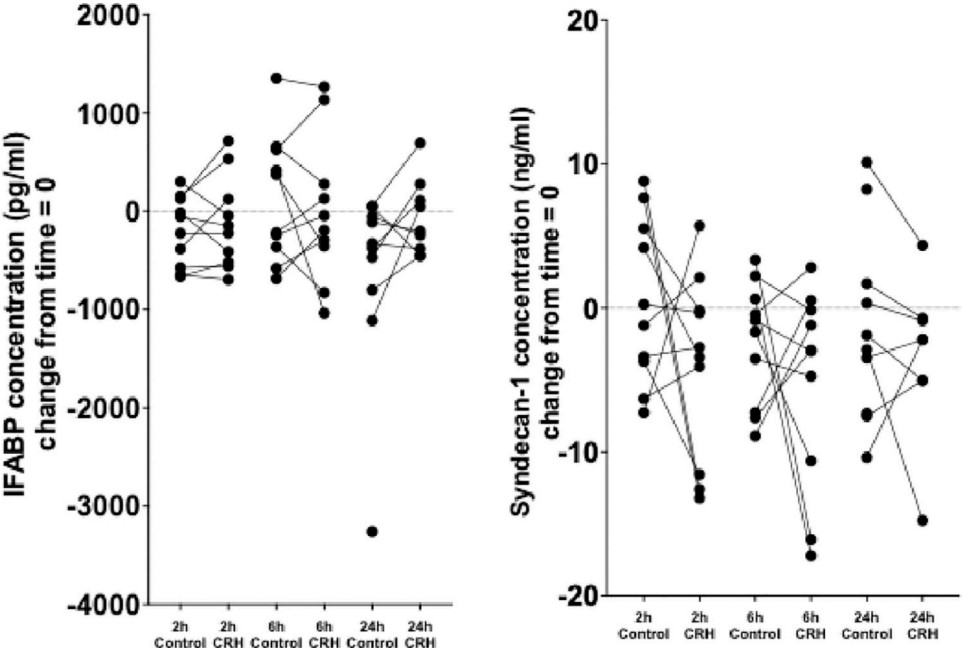

**Fig 5. Changes realtive to baseline (t = 0) in the concentration of biomarkers—intestinal fatty acid-binding protein (IFABP), syndecan-1, lipopolysaccharide-binding protein (LBP), soluble CD14 (sCD14) and LBP:sCD14 ratio—at 2, 6 and 24 hours after saline injection (Control) vs after a high fat high carbohydrate (HFHC) meal.** IFABP levels were lower 6 hours after a HFHC meal intake compared with those following saline injection with a post HFHC IFABP median level of -224.8 (IQR -579.8 to -114.8) pg/ml vs post saline injection of 76.3 (-418 to 634.3) pg/ml (p = 0.04; Wilcoxon), although statistical significance was lost if the data were corrected for multiple comparisons. No other differences for biomarkers at any of the other time points were identified.

There is a paucity of data on the diurnal variations of the circulating intestinal permeability biomarker concentrations, without which interpretation of the effects of interventions is compromised. Because of the reported influence of HFHC meals on some biomarkers, participants were instructed (and partly provided) a 24-hour period of a low-calorie dietary intake. Variations were observed for IFABP. First, there was an overall and consistent reduction in plasma IFABP levels between baseline and 30 minutes prior to baseline measurements. Second, a reduced IFABP level was observed over the entire 24 hours. Such observations were consistent with a study of 8 healthy subjects where a meal intake, whether high or low in fat, reduced IFABP concentrations for 4 hours [29]. In the current study, most participants consumed a light breakfast prior to the study, as these food-related observations were not available during planning for the present study. Hence, oral intake, regardless of its caloric, carbohydrate or fat content, appears to undermine IFABP's effectiveness as a marker of intestinal permeability in the unfasted state. In this context, plasma IFABP likely acts as a reflection of intracellular metabolic processes rather than being indicative of a change in intestinal permeability [21]. The tendency observed for reduction in plasma IFABP observed after the heavier HFHC meal is supportive of this concept. The reduction of plasma IFABP after 24 hours of a low-calorie diet, but not after the HFHC meal is an interesting observation, which also likely reflects IFABP intracellular processes occurring in the setting of a low-calorie but not high-calorie diet the day prior. Therefore, in future studies, plasma IFABP is best measured in the fasting state where the person has had a normal caloric intake in the preceding 24 hours. The variance observed should be taken into account if conditions cannot be well controlled. Fortunately, syndecan-1 and markers of LPS translocation did not display diurnal variation under the conditions studied and were stable when assessed over 30 minutes.

The significant increase in plasma IFABP after CRH injection reported previously [18] could not be clearly replicated in this study. This may reflect the fact that the participants in the present study were allowed a light breakfast, with the post-prandial reduction in their plasma IFABP levels evoking a comparatively smaller IFABP leak from the intestinal epithelium in response to systemic CRH administration. The confounding effect of food ingestion on individuals and variability in the response to food is likely to produce greater variation in IFABP levels after an intervention, an effect that requires greater number of participants to overcome.

More consistency with effects on LPB were observed. The reduction in concentrations of LBP at 2 hours after CRH administration supports the 'mopping up' hypothesis of the circulating plasma LBP by LPS translocation induced with CRH administration. This is in keeping with the presently understood dynamics of LBP concentrations after acute intestinal barrier perturbation in a more robust model (children with intestinal ischaemia after cardiopulmonary bypass surgery) of LPS translocation induction [30]. The fact that, unlike in this robust model, there was no increase in plasma LBP at 24 hours [30] likely reflects the small and transient degree of LPS translocation induced by CRH administration. In contrast, no change to the markers of LPS translocation were observed after the HFHC meal acutely. Therefore, results from the present study do not support previous literature suggesting intestinal barrier perturbation and an increase in LPS translocation in response to a HFHC diet [12–16, 26, 27]. These studies generally showed an increase in measured LPS in the blood, but were unable to show a positive LBP response to greater LPS exposure [12, 13, 15, 16]. Indeed, dual-sugar permeability testing was unable to confirm actual changes in permeability after HFHC meals [16, 31]. Furthermore, the reported acute increase in markers of LPS translocation [12, 13] is contrary to the presently understood dynamics of LBP after epithelial barrier perturbation [30].

The LBP:sCD14 ratio has previously been advocated as a measure of acute LPS translocation [15]. What is currently understood regarding the physiology of LBP and sCD14 dynamics

in response to LPS translocation does not support this proposition and the lack of acute change in this ratio after CRH in the present study further refutes the role of the LBP:sCD14 ratio as a valid marker of acute LPS translocation. Taken together, the overall findings from markers of LPS translocation examined in the present study support the use of intravenous CRH administration in inducing transient LPS translocation and do not support changes to LPS translocation in response to a HFHC meal, either acutely or after 24 hours.

This study presents several key implications for future research in the intestinal permeability field. First, plasma IFABP levels in any individual person are not stable over a 24-hour period. Second, in order to increase its sensitivity to change, plasma IFABP measurements should be performed in the fasted state and a standard caloric intake must be ensured the day prior to measurement. These factors need to be taken into account when powering for and interpreting studies using IFABP. Previous studies linking disease activity of various gastrointestinal conditions to plasma IFABP levels generally did not consider fasting states or the diurnal schedule of their participants. These are factors which, given the results from the present study, would have implications for previous apparently negative findings. However, consistent and/or large changes when IFABP is assessed under similar conditions may still be informative provided appropriate control conditions were met. These diurnal and fasting issues do not seem to be a factor in the other proposed biomarkers of intestinal permeability examined in this study.

Secondly, it appears that parenteral injection of CRH remains a reasonable model for the induction of acute intestinal barrier perturbation. The effect size of this intervention, however, is likely to be small. None the less, this is a convenient model which could be used in future studies aiming to assess the effect of interventions on intestinal barrier dysfunction. Lastly, the present study refutes the ability of the HFHC meal to induce intestinal barrier dysfunction in healthy volunteers. Future research in this field should avoid using this as a model of acute intestinal barrier perturbation.

The present study has limitations that must be addressed. First, owing to the multiple visits, the intensive study protocol and limitations imposed by COVID-19-asssociated restrictions, the study population was small. Further, because this study was intended as a pilot study, no formal power calculation was performed. However, given its exloparory nature and the number of participants recruited for previous similar studies [17, 26], 10 participants who acted as their own controls was considered to be sufficient to elicit signals in intestinal permeability changes associated with present interventions. Carry-over effects from the cross-over design were not apparent. Secondly, the participants were not fasted prior to the baseline blood tests or during the initial 6-hour blood collection, which, as discussed above, is an issue for plasma IFABP level measurements. However, it would be unrealistic and un-physiological to have fasted participants for the entire 24-hour period to establish the fasting diurnal variation of IFABP levels. Therefore, future studies should consider an overnight fast before taking blood samples for plasma IFABP levels and should, for acute interventional studies, consider the confounding variable associated with food intake in the powering of the study. Thirdly, there was no comparator meal–aside from the low fat, low carbohydrate snacks allowed throughout the duration of the study protocol–to enhance interpretation of the effects of a HFHC meal. Previous studies that have shown changes to markers of LPS translocations have included a comparator meal to the HFHC meal challenge. However, one would expect the degree of difference to be greater in the present study as the nutritional content differed to a larger degree than comparator meals in previous studies; yet no acute change to markers of LPS translocation could be detected. Despite its small sample size, this study provides guidance in designing and powering larger studies utilise plasma markers of intestinal permeability. Nevertheless, the findings need to be re-examined in a larger study.

In conclusion, the present study has established that plasma IFABP levels display significant diurnal variation, which is at least partly related to food intake, whereas other examined biomarkers were stable over time. To reduce the potentially confounding effect of food intake on plasma IFABP levels, a fasting sample should be taken, whereas food intake does not alter plasma LBP, sCD14 and syndecan-1 levels. The use of parenteral CRH is a reasonable model to study the effects of mild, acute intestinal barrier perturbation, but its effect on plasma IFABP levels requires repeating in the fasting state. Results from this study do not support the use of the HFHC meal as a model for intestinal barrier perturbation. They also do not support the use of the LBP:sCD14 ratio as a marker of acute intestinal barrier dysfunction and LPS translocation.

## Supporting information

**S1 Checklist. CONSORT 2010 checklist of information to include when reporting a randomised trial\*.**
(DOC)

**S1 Fig. CONSORT 2010 flow diagram.**
(DOC)

**S1 File.**
(DOCX)

**S1 Data.**
(XLSX)

**S2 Data.**
(XLSX)

## Author Contributions

**Conceptualization:** Tamara Mogilevski, Anke Nguyen, Jessica Fitzpatrick, Emma P. Halmos, Peter R. Gibson.

**Data curation:** Tamara Mogilevski, Anke Nguyen.

**Formal analysis:** Tamara Mogilevski, Sam Rosella, Francis Parker.

**Investigation:** Tamara Mogilevski, Emma P. Halmos.

**Methodology:** Tamara Mogilevski, Jessica Fitzpatrick, Emma P. Halmos, Peter R. Gibson.

**Project administration:** Tamara Mogilevski, Sam Rosella, Anke Nguyen.

**Software:** Francis Parker.

**Supervision:** Emma P. Halmos, Peter R. Gibson.

**Writing – original draft:** Tamara Mogilevski, Peter R. Gibson.

**Writing – review & editing:** Tamara Mogilevski, Sam Rosella, Anke Nguyen, Jessica Fitzpatrick, Francis Parker, Emma P. Halmos, Peter R. Gibson.

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
