## [Decision Letter · Decision Letter 0]

9 Aug 2023

PONE-D-23-16375TIME-DEPENDENT CHARACTERISATION OF BIOMARKERS OF INTESTINAL BARRIER FUNCTIONIN RESPONSE TO A HIGH-FAT, HIGH-CARBOHYDRATE MEAL AND CORTICOTROPIN RELEASING HORMONE – A RANDOMISED, PLACEBO-CONTROLLED CROSS-OVER STUDY IN HEALTHY HUMANSPLOS ONE

Dear Dr. Mogilevski,

Thank you for submitting your manuscript to PLOS ONE. After careful consideration, we feel that it has merit but does not fully meet PLOS ONE’s publication criteria as it currently stands. Therefore, we invite you to submit a revised version of the manuscript that addresses the points raised during the review process.

We look forward to receiving your revised manuscript.

Kind regards,

David Chibuike Ikwuka, Ph.D.

Academic Editor

PLOS ONE

Journal Requirements:

2. Thank you for submitting your clinical trial to PLOS ONE and for providing the name of the registry and the registration number. The information in the registry entry suggests that your trial was registered after patient recruitment began. PLOS ONE strongly encourages authors to register all trials before recruiting the first participant in a study.

1) your reasons for your delay in registering this study (after enrolment of participants started);

2) confirmation that all related trials are registered by stating: “The authors confirm that all ongoing and related trials for this drug/intervention are registered”.

Additional Editor Comments: After careful evaluation by our expert reviewers, I am pleased to inform you that your manuscript has been positively received. However, there are a few minor revisions that need to be addressed before we can proceed with the publication process: provide proper description of  the characteristics of study participants in addition to addressing the comments raised by each reviewer.

Reviewers' comments:

Reviewer's Responses to Questions

**Comments to the Author**

1. Is the manuscript technically sound, and do the data support the conclusions?

Reviewer #1: Yes

Reviewer #2: Yes

Reviewer #3: Yes

Reviewer #4: Partly

2. Has the statistical analysis been performed appropriately and rigorously? 

Reviewer #1: Yes

Reviewer #2: Yes

Reviewer #3: Yes

Reviewer #4: Yes

3. Have the authors made all data underlying the findings in their manuscript fully available?

Reviewer #1: Yes

Reviewer #2: Yes

Reviewer #3: Yes

Reviewer #4: Yes

4. Is the manuscript presented in an intelligible fashion and written in standard English?

Reviewer #1: Yes

Reviewer #2: Yes

Reviewer #3: Yes

Reviewer #4: Yes

5. Review Comments to the Author

Reviewer #1: This is a well-written report on a very small study of plasma concentrations of CORTICOTROPIN RELEASING HORMONE. This claims to be a "pilot" or "exploratory" clinical trial that shows that concentrations are not stable. The study does not consider power, and claim that "10 participants who acted as their

own controls was considered to be sufficient to elicit signals in intestinal permeability changes

associated with present interventions." I don't know what "elicit signals" means statistically, but it sounds like the authors are claiming adequate power to detect intestinal permeability changes. This needs to be clarified carefully in a statistical manner, because a low powered study, even if it achieves statistical significance in some metric, is not sufficient for statistical validity. Please clarify.

Second, an "exploratory" or "pilot" study indicates that something will follow. What are the implications for future studies? That is the role of pilot studies, not to definitively show an outcome.

The CONSORT document requires stating the randomization procedure used (blocks with block size, complete (simple randomization, etc.) Patients "were randomised (www.randomisation.com) to receive either a CRH or placebo injection" is not a randomization procedure.

Statistical analysis is adequate. No comments there.

Reviewer #2: Comments: This is an amazing work. But there are some area that should be re-visited.

1.The Sample size is Scanty, more information would have been obtained with a bigger sample size statistically.

2. Gender population, 50% female I believe the biomarker would have had a different effects on different gender when compared, may be comparing the female and the male genders with regards to the biomarker.

3.Sample Containers for specimen collection, EDTA was used for both analyte with protein back ground such as LBP and IFABP assay, Check the appropriate containers for the sample collections.

.

Reviewer #3: REVIEWER’S REPORT TO AUTHORS FOR PONE-D-23-16375

Dear Authors,

The following changes are recommended to be changed after review of the Article titled: “Time-dependent characterization of biomarkers of intestinal barrier function in response to a high-fat, high-carbohydrate meal and corticotropin releasing hormone – a randomized, placebo-controlled cross-over study in healthy humans”

Title: I suggest that the title should be modified to read:

Characterization of biomarkers of intestinal barrier function in response to high-fat/high carbohydrate meal and corticotropin releasing hormone in healthy humans. (20 words title). The randomized placebo-controlled study can be indicated as the type of study in materials and methods.

Abstract: Only A should be capitalized in abstract, use bold type, 18pt font as indicated in the manual style.

Line 40: Add released before over (sentence incomplete)

Line 42-44: Rephrase to- This study aimed to examine the stability and pharmacokinetics of changes in plasma concentrations of intestinal Fatty-acid -binding -protein (IFABP), Lipopolysaccharide-binging –protein (LBP), soluble CD14, and Syndecan-1 after acute stress and high fat-high-carbohydrate meal.

Line 49: Mention sample size used for each group after the last word interventions

Line 61: Introduction, Adhere to body formatting guidelines

Line 91: Add – were after syndecan-1

Line 95-97: Add “but’ before the time line. Rephrase, therefore, can be rapidly released on intestinal injury. The timeline……

Line 99: Sentence in lines 99-105 too long, use; to separate aims

Line 111: Male or female aged between ….remove and

In participants, be specific; male and female? How many recruits from each groups/gender? State your sample size, location of study sites, ethical approval, state study design/type, sampling technique? How were they prepared? Fasting or non-fasting? Which meal did they take the previous day and at what Time?

Study Protocols: Rewrite line 119 (After the assessment by the study doctor to ensure eligibility, informed consent was signed by the participants. Use “Figure 1”

Line 128-129: What volume of normal saline?) They then had further blood tests after 2 and 6 hours during which they remained near the department.in line 129

Line 129 Rephrase: Further blood tests was done after 2 and 6 hours as they remained nearby the sampling site.

Line 139: First visit

Study protocol: Specify contents explained to them as LFLC, HFHC etc. from the beginning of the study or was the above mentioned calories the same as used in the beginning of the study?), in study protocol

Line 172-176: This should be moved to the beginning of study protocols for clarity and simply indicate statistical tools and types used. Example, Wilcoxon ranked sign test etc. as stated below. Timepoints should be spaced as time points

Statistical analysis: Lines 180-182 (Time points) should be spaced out. Clarity about the statistical tools used needed.

Line 196: Were females(Plural)

Lines 204-208: Time points, see lines 204, 208 correct throughout

Line 211: “Figure 3” ; Always represent figure in parentheses, see others.

Line 252- this marked point should go to type of study in methodology. In this study.

Line 280; Un-fasted state

Line 282: Remove observed highlighted. Repeated words

Line 302: Use the sentence, this is in line with

Line 327: Rephrase to and a standard caloric intake must be ensured the day prior to measurement.

References: Abide by the Journal manual reference style. Ensure in all, the recommended style is followed.

Additional Information: Authors should add the tables and figures as attachments. Authors are required to make all data underlying the findings described fully available, without restriction

Authors should enter a financial disclosure statement that describes the sources of funding for the work included in this submission. Review the submission guidelines for detailed requirements. View published research articles from PLOS ONE for specific examples. This statement is required for submission and will appear in the published article if the submission is accepted. Please make sure it is accurate

Unfunded studies Enter: The author(s) received no specific funding for this work. Funded studies Enter a statement with the following details: Initials of the authors who received each award • • Grant numbers awarded to each author • The full name of each funder • URL of each funder website Did the sponsors or funders play any role in the study design, data collection and analysis, decision to publish, or preparation of the manuscript? • NO - Include this sentence at the end of your statement: The funders had no role in study design, data collection and analysis, decision to publish, or preparation.

NB: Please provide the Figures and tables for the work .

.

Reviewer #4: -The method and the results do not clearly address the first aim/ objective of the study,

-The diurnal variation of IFABP over the 24hrs which was one of the key findings was not properly described

-The study had low sample size and the age difference of the participants were wide apart. this may have affected the results

-The cross study design may not be appropriate in this type of study, as the effect of first intervention may affect the second intervention

The author should state clearly how long it took for the study, this will help to determine the level of compliance of the participants and validity of results

-The results were presented at different P values which may have affected the interpretation of the results even though they were corrected

-The figures were not self explanatory, another format of line diagram that clearly shows the levels of the biomarkers across the 24hrs may be better

- The author should clearly state the basis for the time point interval

-The author should objectively discuss the results obtained from the study

-The use of ambiguous statement in result presentation like" There was a tendency for reduction inn plasma LBP at 2 hours after CRH injection compared with that after the saline injection ......" should be revised

-some other grammatical errors should also be corrected

6. PLOS authors have the option to publish the peer review history of their article (what does this mean?). If published, this will include your full peer review and any attached files.

Reviewer #1: No

Reviewer #2: No

Reviewer #3: No

Reviewer #4: **Yes: **Olufunke Onaadepo

---

## [Author Response · Author response to Decision Letter 0]

7 Oct 2023

Reply to reviewer

We thank the reviewer for their comments and made have the following comments and changes to the manuscript.

Reviewer #1: This is a well-written report on a very small study of plasma concentrations of CORTICOTROPIN RELEASING HORMONE. This claims to be a "pilot" or "exploratory" clinical trial that shows that concentrations are not stable. The study does not consider power, and claim that "10 participants who acted as their own controls was considered to be sufficient to elicit signals in intestinal permeability changes associated with present interventions." I don't know what "elicit signals" means statistically, but it sounds like the authors are claiming adequate power to detect intestinal permeability changes. This needs to be clarified carefully in a statistical manner, because a low powered study, even if it achieves statistical significance in some metric, is not sufficient for statistical validity. Please clarify.

Thank you for this feedback. The sample size was chosen based on previously published dietary studies in the intestinal permeability field which also used small numbers of participants. The study was designed to explore if a signal could be detected in the changes to intestinal permeability utilising the two approaches of intestinal barrier perturbation. This point has been clarified in the manuscript in the participants section but was also previously addressed in the manuscript – in the discussion section.

Second, an "exploratory" or "pilot" study indicates that something will follow. What are the implications for future studies? That is the role of pilot studies, not to definitively show an outcome.

Thank you for this feedback. The discussion section has been amended to add in this important point – “Despite its small sample size, this study provides guidance in designing larger studies utilise plasma markers of intestinal permeability.”

The CONSORT document requires stating the randomization procedure used (blocks with block size, complete (simple randomization, etc.) Patients "were randomised (www.randomisation.com) to receive either a CRH or placebo injection" is not a randomization procedure.

Thank you for this feedback. The paper has now been amended to state: “randomised to receive either a CRH or placebo injection first. This was calculated by using randomly computed blocks with random block sizes generated by the website www.randomisation.com.”

Statistical analysis is adequate. No comments there.

Reviewer #2: Comments: This is an amazing work. But there are some area that should be re-visited.

1.The Sample size is Scanty, more information would have been obtained with a bigger sample size statistically.

Thank you for this comment. We agree; however, the significant face-to-face involvement required by participants in the study in the setting of the pandemic precluded a larger number of participants. This limitation is discussed at length in the discussion.

2. Gender population, 50% female I believe the biomarker would have had a different effects on different gender when compared, may be comparing the female and the male genders with regards to the biomarker.

Thank you for this feedback. This was analysed for in the preliminary analysis with no differences between sexes detected. However, as above, we acknowledge the limitation of small participant numbers. Especially when looking at subgroup analyses. 

3.Sample Containers for specimen collection, EDTA was used for both analyte with protein back ground such as LBP and IFABP assay, Check the appropriate containers for the sample collections.

Thank for this feedback. The samples were collected into contains which had previously been used for this analysis. A reference has been added to the methods section for clarity.

.

Reviewer #3: REVIEWER’S REPORT TO AUTHORS FOR PONE-D-23-16375

Dear Authors,

The following changes are recommended to be changed after review of the Article titled: “Time-dependent characterization of biomarkers of intestinal barrier function in response to a high-fat, high-carbohydrate meal and corticotropin releasing hormone – a randomized, placebo-controlled cross-over study in healthy humans”

Title: I suggest that the title should be modified to read:

Characterization of biomarkers of intestinal barrier function in response to high-fat/high carbohydrate meal and corticotropin releasing hormone in healthy humans. (20 words title). The randomized placebo-controlled study can be indicated as the type of study in materials and methods. 

This has been amended as suggested, thank you.

Abstract: Only A should be capitalized in abstract, use bold type, 18pt font as indicated in the manual style.

This has been corrected, thank you

Line 40: Add released before over (sentence incomplete)

I am not clear which sentence the reviewer is referring to. The sentence on the line 2 above the sentence in the comment below appears complete. 

Line 42-44: Rephrase to- This study aimed to examine the stability and pharmacokinetics of changes in plasma concentrations of intestinal Fatty-acid -binding -protein (IFABP), Lipopolysaccharide-binging –protein (LBP), soluble CD14, and Syndecan-1 after acute stress and high fat-high-carbohydrate meal.

This has been amended – thank you for the wording suggestion.

Line 49: Mention sample size used for each group after the last word interventions

The same sample size was used for each intervention – this paragraph has been amended to clarify this point as the original wording was indeed confusing.

Line 61: Introduction, Adhere to body formatting guidelines

Corrected, thank you

Line 91: Add – were after syndecan-1

We are sorry, but do not understand this requested change.

Line 95-97: Add “but’ before the time line. Rephrase, therefore, can be rapidly released on intestinal injury. The timeline……

Thank you for drawing our attention to this problematic sentence – this has now been rephrased. 

Line 99: Sentence in lines 99-105 too long, use; to separate aims

Amended as requested

Line 111: Male or female aged between ….remove and

Amended as requested

In participants, be specific; male and female? How many recruits from each groups/gender? State your sample size, location of study sites, ethical approval, state study design/type, sampling technique? How were they prepared? Fasting or non-fasting? Which meal did they take the previous day and at what Time?

Thank you – the participant question was addressed in the protocol. Ethics approval, study location sites, sampling technique, study design type, fasting and oral intake status are all addressed in the study protocol section of the manuscript.

Study Protocols: Rewrite line 119 (After the assessment by the study doctor to ensure eligibility, informed consent was signed by the participants. Use “Figure 1”

Thank you. This has been corrected.

Line 128-129: What volume of normal saline?) They then had further blood tests after 2 and 6 hours during which they remained near the department.in line 129

Thank you. This was 10ml and has been clarified in the manuscript.

Line 129 Rephrase: Further blood tests was done after 2 and 6 hours as they remained nearby the sampling site.

Thank you for the suggestion – this has been amended.

Line 139: First visit

Study protocol: Specify contents explained to them as LFLC, HFHC etc. from the beginning of the study or was the above mentioned calories the same as used in the beginning of the study?), in study protocol

Thank you. Line 142 of the revised manuscript states regarding the LFLC diet: These were the only foods that they were able to consume during the study

Line 172-176: This should be moved to the beginning of study protocols for clarity and simply indicate statistical tools and types used. Example, Wilcoxon ranked sign test etc. as stated below. Timepoints should be spaced as time points

Thank you for this feedback, however we feel that the clinical aspect of the trial rather should be specified prior to the laboratory specifications so have left the order as such in the manuscript. The statistical analysis is specified in line 204 of the new manuscript as suggested in this comment.

Statistical analysis: Lines 180-182 (Time points) should be spaced out. Clarity about the statistical tools used needed.

Thank you for this feedback. The statistical methods are described in detail in lines 212 to 225 of the revised manuscript.

Line 196: Were females(Plural)

Thank you for this feedback. We feel ‘one half were female’ rather than ‘one half were females’ may read nicer. 

Lines 204-208: Time points, see lines 204, 208 correct throughout

Thank you. This has been amended in the manuscript as suggested. 

Line 211: “Figure 3” ; Always represent figure in parentheses, see others.

Thank you. This has been amended in the manuscript. 

Line 252- this marked point should go to type of study in methodology. In this study.

Line 280; Un-fasted state

Line 282: Remove observed highlighted. Repeated words

Line 302: Use the sentence, this is in line with

Line 327: Rephrase to and a standard caloric intake must be ensured the day prior to measurement.

Thank you for this feedback. Most of these points have been amended in the manuscript.

References: Abide by the Journal manual reference style. Ensure in all, the recommended style is followed.

Additional Information: Authors should add the tables and figures as attachments. Authors are required to make all data underlying the findings described fully available, without restriction

Thank you, this has been revised in this format.

Authors should enter a financial disclosure statement that describes the sources of funding for the work included in this submission. Review the submission guidelines for detailed requirements. View published research articles from PLOS ONE for specific examples. This statement is required for submission and will appear in the published article if the submission is accepted. Please make sure it is accurate

Unfunded studies Enter: The author(s) received no specific funding for this work. Funded studies Enter a statement with the following details: Initials of the authors who received each award • • Grant numbers awarded to each author • The full name of each funder • URL of each funder website Did the sponsors or funders play any role in the study design, data collection and analysis, decision to publish, or preparation of the manuscript? • NO - Include this sentence at the end of your statement: The funders had no role in study design, data collection and analysis, decision to publish, or preparation.

NB: Please provide the Figures and tables for the work .

.

Reviewer #4: -The method and the results do not clearly address the first aim/ objective of the study,

-The diurnal variation of IFABP over the 24hrs which was one of the key findings was not properly described

-The study had low sample size and the age difference of the participants were wide apart. this may have affected the results

Thank you for this feedback – the issue of the small sample size has been addressed in the manuscript. 

-The cross study design may not be appropriate in this type of study, as the effect of first intervention may affect the second intervention

Thank you for this feedback. We feel the washout period of 2 weeks a significant duration of time to wash out any effect of corticotropin releasing hormone. The half-life of this is less than 10 minutes and the intestinal barrier can replenish itself in under 1 week. Analysis of the responses to CRH were similar whether it or saline were used first. This is now included in the results.

The author should state clearly how long it took for the study, this will help to determine the level of compliance of the participants and validity of results

Thank you for this feedback. The times taken for each arm and visit of the study are presented in “figure 1” - flowchart of the study. 

-The results were presented at different P values which may have affected the interpretation of the results even though they were corrected

Thank you for this feedback – the correction was felt necessary by our statistician and previous reviewers. 

-The figures were not self explanatory, another format of line diagram that clearly shows the levels of the biomarkers across the 24hrs may be better

Thank you for this feedback. The legends have been expanded. The selected time points were chosen to draw attention to the specific time points. We had previously expressed the markers across a 24-hour time frame but the graphs lacked clarity when presented in such a manner.

- The author should clearly state the basis for the time point interval

Thank you – this has been amended in the revised manuscript. 

-The author should objectively discuss the results obtained from the study

-The use of ambiguous statement in result presentation like" There was a tendency for reduction inn plasma LBP at 2 hours after CRH injection compared with that after the saline injection ......" should be revised

The results are now objectively described.

-some other grammatical errors should also be corrected

---

## [Decision Letter · Decision Letter 1]

13 Nov 2023

CHARACTERISATION OF BIOMARKERS OF INTESTINAL BARRIER FUNCTION IN RESPONSE TO A HIGH FAT/HIGH CARBOHYDRATE MEAL AND CORTICOTROPIN RELEASING HORMONE.

PONE-D-23-16375R1

Dear Dr. Mogilevski,

We’re pleased to inform you that your manuscript has been judged scientifically suitable for publication and will be formally accepted for publication once it meets all outstanding technical requirements.

Kind regards,

David Chibuike Ikwuka, Ph.D.

Academic Editor

PLOS ONE

Additional Editor Comments (optional):

Reviewers' comments:

Reviewer's Responses to Questions

**Comments to the Author**

1. If the authors have adequately addressed your comments raised in a previous round of review and you feel that this manuscript is now acceptable for publication, you may indicate that here to bypass the “Comments to the Author” section, enter your conflict of interest statement in the “Confidential to Editor” section, and submit your "Accept" recommendation.

Reviewer #5: All comments have been addressed

2. Is the manuscript technically sound, and do the data support the conclusions?

Reviewer #5: Yes

3. Has the statistical analysis been performed appropriately and rigorously? 

Reviewer #5: Yes

4. Have the authors made all data underlying the findings in their manuscript fully available?

Reviewer #5: Yes

5. Is the manuscript presented in an intelligible fashion and written in standard English?

Reviewer #5: Yes

6. Review Comments to the Author

Reviewer #5: I appreciate the opportunity to examine this fascinating work.

As remarked by the other reviewers, it would have been beneficial to do a formal power analysis to ascertain whether the sample size is sufficient to detect changes in intestinal permeability. The researchers acknowledge this fact in their manuscript. It should be noted that numerous factors influence the power analysis, one of which is measurement variability (standard deviation). A formal power analysis may be impossible because this is an exploratory study, and variability information may not be available. Furthermore, the researchers acknowledged the limits imposed by a limited sample size, and they used a pragmatic approach in selecting the chosen subjects and used relevant statistical tests (ANOVA, Friedman, and Wilcoxon signed rank test) and an adequate statistical significance threshold of 5%. The researchers have shown that these interventions will work in typical clinical care. Further, the researchers identify critical implications of this study for future research. Based on these points and the fact that the authors have adequately addressed all the reviewers’ comments, I have no hesitation in recommending this work for publication.

Thank you.

7. PLOS authors have the option to publish the peer review history of their article (what does this mean?). If published, this will include your full peer review and any attached files.

Reviewer #5: No

---

## [Editor Report · Acceptance letter]

13 Feb 2024

PONE-D-23-16375R1 

PLOS ONE

Dear Dr. Mogilevski, 

I'm pleased to inform you that your manuscript has been deemed suitable for publication in PLOS ONE. Congratulations! Your manuscript is now being handed over to our production team.

Kind regards, 

on behalf of

Dr David Chibuike Ikwuka 

Academic Editor

PLOS ONE